# Plasmidome, resistome, and virulence-associated gene characterization of *Acinetobacter johnsonii* in NASA cleanrooms and a clinical setting

Anna Tumeo,[1] Georgios Miliotis,[1] Andy O'Connor,[1] Varsha Vijayakumar,[2] Pratyay Sengupta,[2,3,4] Francesca McDonagh,[1] Aneta Kovarova,[1] Christina Clarke,[5] Brigid Hooban,[6] Nitin Kumar Singh,[7] Alexandre Soares Rosado,[8] Karthik Raman,[3,9] Kasthuri Venkateswaran[7]

**ABSTRACT**   Evidence suggests the persistence of non-spore-forming *Acinetobacter johnsonii* in high-stakes controlled and nutrient-limited environments. Here, we investigated the mechanisms underlying this adaptability through a comprehensive genomic analysis of 22 isolates of *A. johnsonii* from NASA's Payload Hazardous Servicing Facility (PHSF) and one carbapenem-resistant strain (E154408A) from patient colonization in Ireland. Core-genome phylogeny revealed clustering of PHSF-originating isolates in a monophyletic clade divergent from the main species lineage. Species-wide virulence-associated genes and metabolic reconstruction indicated the exclusive presence in PHSF-originating isolates of two complete efflux pumps and a conserved allantoin racemase, suggesting adaptability for multiple environmental stresses. The ubiquity of $bla_{OXA}$ in genomes analyzed ($n = 112$) and the phenotypically validated multidrug-resistant profile of the E154408A strain highlight *A. johnsonii*'s potential as an antimicrobial resistance (AMR) reservoir. Plasmidome analysis suggested gain/loss events across the monophyletic population and potential AMR acquisition pathways. Genome-to-metagenome mapping identified genomic signatures of *A. johnsonii* in PHSF >10 years post-initial isolation.

**IMPORTANCE**   *Acinetobacter johnsonii* is increasingly recognized as an emerging human pathogen, with growing evidence of its ability to persist in controlled, high-stakes environments, posing risks as both a persistent environmental contaminant and an antimicrobial resistance (AMR) reservoir. Yet, gaps remain in our understanding of its AMR profile and the mechanisms that enable its enhanced environmental adaptability. This knowledge is necessary in contexts where biological cleanliness is a priority, such as clinical settings and spacecraft assembly facilities' cleanrooms, where contamination of hardware with terrestrial microorganisms is concerning. In this study, we aim to address some of the key knowledge gaps by providing genomic insights into a rare multidrug-resistant clinical isolate and 22 NASA cleanroom isolates that persisted for over a decade in extremely clean conditions. Our findings will help assess the contamination risk of *A. johnsonii* in high-stakes environments and ultimately strengthen our ability to manage this microbial contaminant across terrestrial and extraterrestrial settings.

## HIGHLIGHTS

- Cleanroom-derived *A. johnsonii* genomes show traits consistent with increased adaptability.
- Genomic signatures of *A. johnsonii* persisted in the cleanrooms for over 10 years.
- $bla_{OXA}$ is ubiquitously found in all 112 *A. johnsonii* genomes analyzed.

**Peer Reviewer** Phoebe Lostroh, Colorado College, Colorado Springs, Colorado, USA

Address correspondence to Georgios Miliotis, georgios.miliotis@universityofgalway.ie, or Karthik Raman, kraman@dsai.iitm.ac.in.

Anna Tumeo and Georgios Miliotis are joint first authors. Author names are arranged alphabetically according to first name.

The authors declare no conflict of interest.

See the funding table on p. 15.

- Isolate E154408A is the first reported patient colonization case by carbapenem-resistant *A. johnsonii* in Europe.

**KEYWORDS** *Acinetobacter johnsonii*, Payload Hazardous Servicing Facility, NASA, cleanrooms, extremotolerant, AMR, carbapenem-resistant, oxacillinase, efflux pumps, allantoin racemase

The *Acinetobacter* genus is known to thrive in controlled environments, largely due to robust biofilm formation enabling survival in desiccation and nutrient-limited conditions (1) and resistance to cleaning and disinfection efforts (2). In addition, the widespread presence of efflux pumps (e.g., AdeABC) in the genomes of *Acinetobacter* species, as well as their known tendency to acquire antimicrobial resistance genes (ARGs) to clinically relevant antibiotics (e.g., $bla_{OXA}$ beta-lactamases), (3) facilitates their adaptability for multiple environmental stresses and their persistence despite exposure to heavy metals and antibiotic residues (4). *Acinetobacter* species presenting increasingly complex antimicrobial resistance (AMR) profiles, therefore, pose significant clinical challenges when associated with hospital-acquired infections.

*Acinetobacter johnsonii* is a Gram-negative, non-spore-forming, non-fermentative coccobacillus within the *Acinetobacter* genus (5). As a species, it exhibits ecological versatility, commonly isolated from environmental niches, such as agricultural soil, freshwater systems, including rivers and lakes, marine ecosystems, and anthropogenically impacted sites. Although less well understood than *Acinetobacter baumannii*, the clinical relevance of *A. johnsonii* is also recently gaining attention. *A. johnsonii*'s adaptability to desiccation and low-nutrient conditions has indeed been highlighted by its isolation from clinical settings (6) and NASA Spacecraft Assembly Facilities (SAF) cleanrooms (7). A 2020 phylogeographical study identified *A. johnsonii* isolates from both environmental and clinical sources that circulate globally and harbor antibiotic-resistance genes, including carbapenemases, underscoring its clinical relevance (8). *A. johnsonii* has also been implicated in opportunistic infections such as bacteremia, meningitis, and post-traumatic wound infections, often in immunocompromised hosts (9). Furthermore, scarce reports of carbapenemase-producing *A. johnsonii* isolates carrying $bla_{OXA}$ and $bla_{NDM-1}$ highlight its potential as an understudied emerging AMR threat (10).

Critical knowledge gaps remain regarding the genomic adaptations that underpin not only *A. johnsonii*'s role as a potential reservoir or vector of AMR within healthcare settings but also its persistence as a non-spore-forming microorganism in harsh and nutrient-limited environments and despite repeated disinfection protocols, such as those carried out within SAF cleanrooms. Ensuring biological cleanliness while assembling and launching spacecraft is, however, critical for planetary protection, life-detection missions, and contamination of hardware with terrestrial microorganisms is concerning (7).

In this study, we aim to address some of these knowledge gaps by conducting a comprehensive genomic analysis of 22 *A. johnsonii* isolates, isolated during and after the assembly and testing of NASA's Mars Phoenix lander at the Payload Hazardous Servicing Facility (PHSF) in the Kennedy Space Center (KSC, Florida, USA), where they survived rigorous disinfection protocols, as well as strain E154408A, identified as the first reported carbapenem-resistant patient colonization case in Ireland and Europe. Alongside species-wide genomic characterization, we investigate the AMR profiles of PHSF-derived isolates and the E154408A strain and validate their resistance to clinically relevant antibiotics with phenotypic assays. Our findings will help assess the contamination risk of *A. johnsonii* in high-stakes low-biomass environments, inform the development of effective mitigation and antimicrobial stewardship strategies, and ultimately strengthen our ability to manage *A. johnsonii* across both terrestrial and extraterrestrial settings.

## MATERIALS AND METHODS

A graphical visualization of the experimental design of this study is presented as Fig. 1.

### Sample collection and cultivation

Cleanroom microorganisms were isolated using culture-based methods optimized for low-biomass environments (7). Environmental samples were collected from up to 10 locations within the KSC-PHSF using Biological Sampling Kits (BiSKits; QuickSilver Analytics) during assembly and testing of the NASA Mars Phoenix lander (2P; 27 June 2007) and immediately after transfer of the spacecraft to the launch pad (3P; 01 August 2007). Sample aliquots from each location and sampling event were processed immediately and plated on R2A agar and other selective media following stress-enrichment treatments (e.g., heat shock, UV, or oxidant exposure), then incubated under nine distinct conditions to maximize recovery of diverse and resistant taxa (7). Morphologically distinct colonies were purified by restreaking and archived for downstream phenotypic and genomic analyses, resulting in the recovery of multiple isolates, including 22 *A. johnsonii* strains. One additional clinical multidrug-resistant (MDR) isolate of *A. johnsonii* (E154408A strain) was isolated in March 2019 through a rectal swab from a 73-year-old patient at Galway University Hospital during routine screening for carbapenem-resistant organisms by Galway Reference Laboratory Service. The patient did not show any signs of infection. Specimen E154408A was collected using a Polystyrene & Viscose Amies Charcoal Swab, which was inoculated onto CHROMagar mSuperCARBA and incubated at 37°C for 48 hours. Suspect colonies were identified by matrix-assisted laser desorption/ionization time-of-flight (MALDI-TOF) mass spectrometry (Bruker microflex, Bruker Daltonics GmbH, Bremen, Germany) (11). Colonies were then purified by restreaking and archived for downstream phenotypic and genomic analyses.

### DNA sequencing and genome assembly

PHSF-originating *A. johnsonii* isolates ($n = 22$) were sequenced using third-generation DNA sequencing (Oxford Nanopore Technologies, Oxford, UK) with FLO-PRO114M, R10.4.1 technology. Raw reads were quality controlled using FastQC version 0.12. Genomic DNA assembly was conducted using Canu (12) version 2.2 and Flye (13) version 2.9.1. For each genome, the optimal representative assembly was identified using dRep (14) version 3.4.5. *A. johnsonii* E154408A strain was sequenced using second-generation DNA sequencing MiSeq platform (PE150; Illumina, Inc., San Diego, CA, USA). Raw reads were filtered using fastp (15) version 0.23.4 using default settings. Filtered 1,322,156 reads were retained for genome assembly, conducted using Shovill version 1.1.0 (see Table S1 at https://doi.org/10.5281/zenodo.18473520).

### *In vitro* and *in silico* species typing

To provide a broader genomic perspective of the 22 PHSF-derived isolates and E154408A strain, an average nucleotide identity (ANI) cluster map was generated using ANIclustermap, including all NCBI RefSeq- and GenBank-available *A. johnsonii* genomes ($n = 89$), excluding metagenome-assembled genomes ($n = 143$) and anomalous assemblies ($n = 21$) (see Table S2 at https://doi.org/10.5281/zenodo.18473520). Among the PHSF-derived isolates, 3P2-tot-A was selected as a representative isolate based on the 99.9% ANI similarity observed across all PHSF-originating isolates and subjected to detailed genomic species typing. Confirmation of *A. johnsonii* 3P2-tot-A involved (i) ANI comparisons against *A. johnsonii* reference genome ANC 3681 (GenBank: GCA_000368805.1) and type strain CIP 64.6$^T$ (GenBank: GCA_000368045.1); (ii) digital DNA–DNA hybridization (dDDH) using the Genome-to-Genome Distance Calculator (16) version 3.0; and (iii) BLASTN analysis of *gyrB* and 16S rRNA gene against the *A. johnsonii* reference genome and type strain.

Due to the extended time gap between isolation (June 2007) and analysis (2024), only 14 of the original 22 PHSF-derived isolates could be successfully revived from the freezer

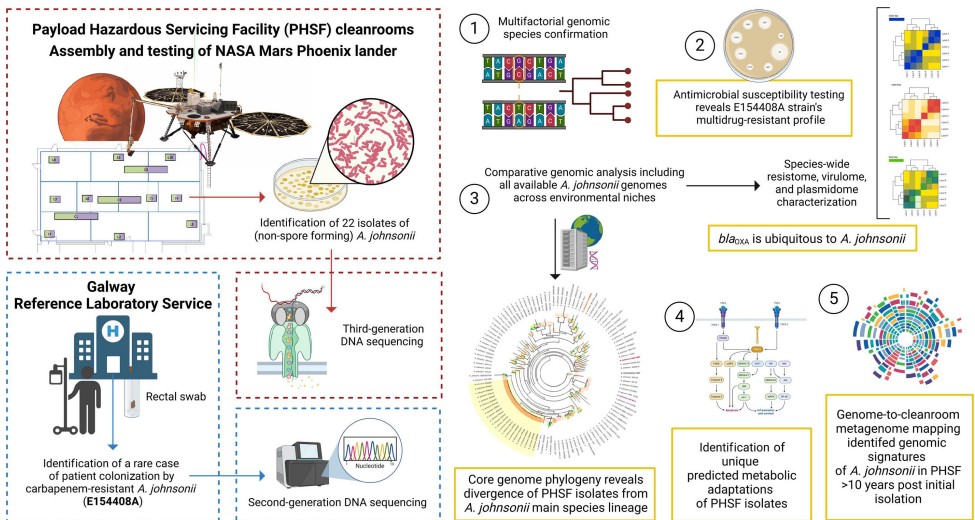

**FIG 1** Experimental design. Graphical visualization of the experimental design, including isolate collection, DNA sequencing, species typing, antimicrobial susceptibility testing, and downstream bioinformatic analysis.

stock for phenotypic species confirmation. In addition to the genome-based phylogeny, taxonomic placement of these recovered isolates, along with E154408A, underwent purity and species confirmatory testing using MALDI-TOF mass spectrometry (Bruker microflex, Bruker Daltonics GmbH, Bremen, Germany) (11).

## Genome annotation and phylogenomic analysis

Assembled genomes were assessed for quality, completeness, and contamination using quast version 5.3.0 and checkM version 1.2.3 (17), respectively. All assembled genomes were annotated using Prokka (18) version 1.14.5. A total of 112 *A. johnsonii* genomes were included in all subsequent analyses, distributed as follows: (i) 89 genomes from NCBI RefSeq and GenBank (all available *A. johnsonii* genomes excluding metagenome-assembled and atypical genomes), (ii) 22 PHSF-derived isolates, and (iii) E154408A. Utilizing these genomes, Roary version 3.13.0 (19) was used to generate the *A. johnsonii* pangenome and define the species core genome. A Maximum Likelihood phylogeny of the PHSF-derived isolates and E154408A strain was constructed based on a multi-sequence alignment of all core genes ($n = 976$) using RAxML version 8.2.13 (20) with the GTRGAMMA model and 1000 bootstraps. The reference genome of the taxonomically adjacent *Acinetobacter haemolyticus* HW-2A (GenBank: GCA_003323815.1) served as an outgroup in the phylogenomic analysis. The ETE toolkit (21) was used for tree manipulation, analysis, and visualization. An SNP analysis was conducted to assess the clonality of PHSF-originating isolates using snipit, selecting *A. johnsonii* 2P07AA as the earliest reference genome.

## Functional annotation of nucleic acid sequences and metabolic mapping

Functional annotation of nucleic acid sequences from all genomes of *A. johnsonii* ($n = 112$) according to KEGG Orthology, Enzyme Commission (EC), and COG category annotations was performed with ggNOG-mapper version 2.1.12 (22) using MMseqs for the search step. Resulting annotated genes were subsequently represented in 232 metabolic maps using KEGGCharter version 1.1.2 (23).

## Species-wide characterization of the resistome, putative virulence-associated genes, plasmidome, and antibacterial biocide and metal resistance gene profile of *A. johnsonii*

All 112 available genomes of *A. johnsonii*, including PHSF-originating isolates and E154408A strain, were screened for the presence of (i) ARGs; (ii) virulence factors (VFs);

and (iii) antibacterial biocide- and metal-resistance genes using ABRicate version 1.0.0 with the CARD 2023 (24), VFDB 2022 (25), and BacMet version 2.0 (26) databases, respectively. Only hits showing a minimum of 80% sequence identity and 60% coverage were retained. A custom database containing all *ade* homologs across *Acinetobacter* species was used to search for Ade efflux-pump-encoding genes. Species-wide plasmidome was characterized using MOB-suite version 3.1.9 (27) with the sub-module MOB-recon version 3.1.9. ABRicate version 1.0.0 was ultimately used on the predicted plasmid sequences to identify plasmidic ARGs.

## Antimicrobial susceptibility testing

Antimicrobial susceptibility testing (AST) was performed on the available PHSF-originating isolates ($n = 14$) and the E154408A strain. Twelve clinically relevant antimicrobials with interpretive criteria under EUCAST and CLSI guidelines were selected for testing: piperacillin (100 µg), ampicillin-sulbactam (10/10 µg), cefepime (30 µg), cefotaxime (30 µg), imipenem (10 µg), meropenem (10 µg), gentamicin (10 µg), tobramycin (10 µg), amikacin (30 µg), doxycycline (30 µg), tetracycline (30 µg), and ciprofloxacin (5 µg). Quality control strains *Escherichia coli* ATCC 25922 and *Pseudomonas aeruginosa* ATCC 27853 were included and maintained at standard conditions (35°C ± 2°C) according to the CLSI and EUCAST guidelines. However, as optimal growth for *A. johnsonii* occurs at temperatures ranging from 15°C to 30°C (28), *A. johnsonii* isolates were incubated at 28°C–30°C. Clinical breakpoints for *Acinetobacter* species defined and established by the CLSI guidelines (CLSI M100-ED34:2024) were applied.

## Estimating the abundance of *A. johnsonii* in NASA cleanrooms

Raw shotgun metagenomic reads from two NASA cleanrooms, namely SAF at Jet Propulsion Laboratory (JPL, California, USA) and PHSF at KSC, and two time points (i.e., 2016 and 2018) were acquired from the NCBI-SRA database (NCBI project accessions: PRJNA1150505 and PRJNA641079). Particularly, the metagenomic reads from 2016 were obtained from SAF at JPL (29), whereas the reads from 2018 were obtained from both cleanrooms (30). The metagenomic data from 2016 included 236 samples, of which 116 were treated with propidium monoazide (PMA) to selectively remove the genomic DNA of dead bacterial targets (31). Data from 2018 included 94 samples, of which 24 PMA-treated and 23 PMA-untreated were from JPL-SAF and KSC-PHSF, respectively. Quality control and data filtering of raw metagenomic reads were performed using fastp version 0.22.0 (15). MetaCompass version 2.0 (32) was used to map filtered metagenomic reads from 2016 and 2018 to *A. johnsonii* 2P07AA, selected as the earliest reference genome, to gain insights into the abundance and prevalence of *A. johnsonii* in JPL and KSC controlled environments. The fraction of mapped reads to the reference genome was used to calculate the corresponding breadth of coverage for PMA-treated and untreated samples from both 2016 and 2018, setting a contig length cut-off score of 1,000 bp.

## RESULTS

### Species-wide ANI comparison

A species-wide all-vs-all ANI comparison of *A. johnsonii* revealed a distinct cluster of the 22 PHSF-derived isolates, each sharing 99.9% ANI (see Fig. S1 at https://doi.org/10.5281/zenodo.18473520). In contrast, PHSF-derived isolates exhibited ANI values ranging from 95.80% to 95.89% when compared to E154408A, confirming species-level relatedness but with notable genomic divergence. Based on the over 99.9% ANI-based relatedness between PHSF-originating genomes, the 3P2-tot-A isolate was selected as a representative for in-depth genomic species typing of the PHSF-derived cluster, alongside E154408A.

## In silico species typing

Over 99% 16S rRNA and *gyr*B sequence identity of 3P2-tot-A isolate to both *A. johnsonii* reference genome ANC 3681 and type strain CIP 64.6[T] supports species-level identity (Table 1). The observed over 95% ANI between the *A. johnsonii* 3P2-tot-A isolate and both ANC 3681 (95.50%) and CIP 64.6[T] (95.83%) strains confirmed species-level relatedness (Table 1). dDDH values > 65% between the 3P2-tot-A isolate and both ANC 3681 (65.60%) and CIP 64.6[T] (66.20%) strains further suggested species-level identity (33). Analogous results were obtained for the E154408A strain (Table 1), confirming that both the PHSF-derived isolates and the E154408A strain taxonomically belong to *A. johnsonii*.

## Estimating the abundance of *A. johnsonii* in NASA cleanrooms

Metagenome mapping with MetaCompass was conducted to estimate the prevalence of *A. johnsonii* in NASA cleanrooms (Fig. 2A through C). In samples from JPL-SAF, 2016, the largest percentage of mapped reads obtained was 6.62% for PMA-untreated and 5.37% for PMA-treated samples, suggesting an overall low yet persisting presence of either viable cells or residual genomic signatures of *A. johnsonii* in the cleanroom. While the breadth of coverage of *A. johnsonii* genome in PMA-treated samples was generally low (average 12.51%) across all locations, higher values (average 27.88%) were measured in PMA-untreated samples with a peak of 76.68% at location 11 (Fig. 2D). A maximum value of 1.92% mapped reads was obtained in PMA-treated samples from JPL-SAF in 2018, suggesting the persistence of a minimal population of viable cells of *A. johnsonii* in the cleanroom (Fig. 2E). However, up to 9.05% reads were mapped in PMA-untreated samples, indicating a significant fraction of the signal coming from non-viable cells. Analogously, low breadth of coverage (average 4.15%; maximum 5.99%) was measured in PMA-treated samples, whereas up to 71.68% coverage (average 25.87%) was obtained for PMA-untreated samples. Similarly, metagenome mapping with samples collected from KSC-PHSF in 2018 yielded up to 1.87% mapped reads with low breadth of coverage (average 4.58%; maximum 11.03%) in PMA-treated samples and up to 13.41% mapped reads with higher coverage (average 79.86%; maximum 89.37%) in PMA-untreated samples (Fig. 2F). Collectively, these results suggest a higher *A. johnsonii* load in the samples collected from KSC-PHSF in 2018 compared to those collected from JPL-SAF in the same year.

## Phylogenomic analysis

The phylogeny of the 22 *A. johnsonii* PHSF-derived isolates and E154408A strain was inferred using a core-genome based maximum-likelihood approach based on all available *A. johnsonii* genomes (*n* = 112) (Fig. 3). The PHSF-derived isolates formed a distinct monophyletic clade differing by 40–77 core SNPs, diverging from the main species lineage. *A. johnsonii* M19 strain (environmental source, Shandong, China) was identified as the closest phylogenetic relative to this cluster, followed by a monophyletic clade comprising *A. johnsonii* strains C4 (environmental source, Cahuita National Park, Costa Rica), JH7 (environmental source, Guangxi, China), GD03955, and GD03761 (both from environmental sources, Pakistan). Meanwhile, *A. johnsonii* E154408A grouped with XY27 strain (animal source, Shanghai, China), GD03727 (environmental source, Pakistan), and mNGS2101_37 (human source, Hangzhou, China).

**TABLE 1** Species typing of strains 3P2-tot-A and E154408A based on ANI, dDDH, 16S rRNA gene, and *gyr*B sequence similarity compared to both *A. johnsonii* reference genome ANC 3681 and *A. johnsonii* type strain CIP 64.6[T]

| Comparison | ANI (%) | dDDH (%) | 16S rRNA sequence identity (%) | *gyrB* sequence identity (%) |
|---|---|---|---|---|
| *A. johnsonii* 3P2-tot-A vs ref. genome ANC 3681 | 95.50 | 65.60 | 99.74 | 98.51 |
| *A. johnsonii* 3P2-tot-A vs type strain CIP 64.6[T] | 95.83 | 66.20 | 99.67 | 97.21 |
| *A. johnsonii* E154408A vs ref. genome ANC 3681 | 95.92 | 66.10 | 99.58 | 97.09 |
| *A. johnsonii* E154408A vs type strain CIP 64.6[T] | 95.81 | 65.70 | 99.65 | 97.13 |

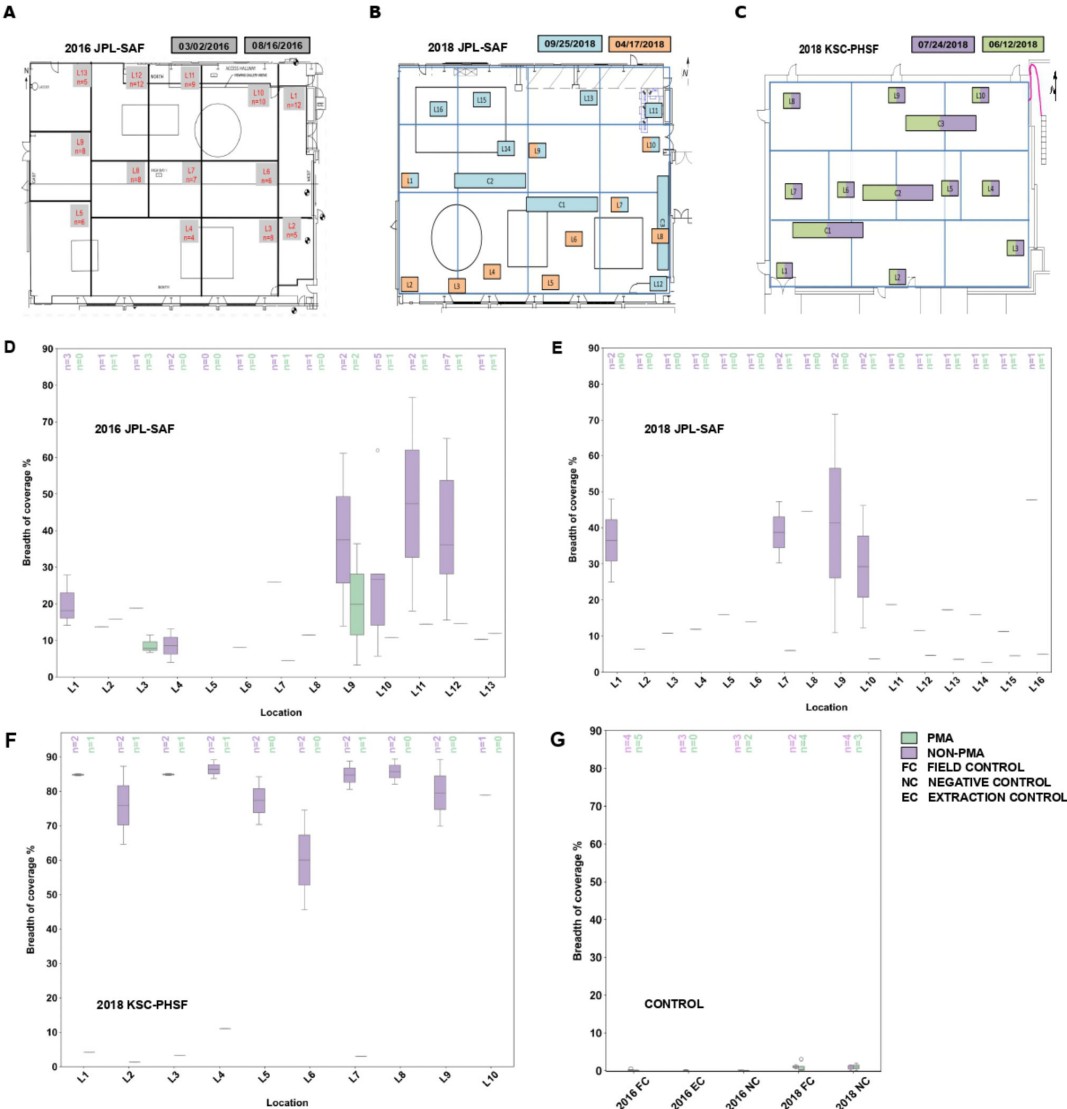

**FIG 2** Estimating the abundance of *A. johnsonii* in NASA cleanrooms. Metagenome mapping results of *A. johnsonii* in cleanrooms. (A–C) Schematic representation of dates and specific locations for samples collected at JPL-SAF over a 6-month period in 2016 (A); at JPL-SAF during two separate events in 2018 (B); and at KSC-PHSF during two separate events in 2018 (C). The colors of the squares correspond to the sampling date. The graph is divided into artificial quadrants based on sample grouping and foot traffic. Dates are presented in the following format: month/date/year. (D–F) Box plots showing the breadth of coverage in percentage for *A. johnsonii* for PMA-treated (PMA) vs PMA-untreated (non-PMA) samples across different sampling locations in 2016 JPL-SAF (D); 2018 JPL-SAF (E); and 2018 KSC-PHSF (F). (G) Box plot depicting the breadth of coverage in percentiles for controls, comparing PMA vs non-PMA for years 2016 and 2018.

## Functional annotation of nucleic acid sequences and metabolic mapping

A total of 3147 different genes across all *A. johnsonii* genomes (*n* = 112) were functionally annotated using eggNOG-mapper. An average of 2,070 (range: 2,052–2,075) genes were annotated in PHSF-originating isolates, one of these coding for an allantoin racemase (KO: K16841; EC: 5.1.99.3; COG4126) involved in purine metabolism, which was exclusively detected in PHSF-originating genomes except for the 2P08AD strain (Fig. 4A). Two thousand fifty genes were annotated in the genome of *A. johnsonii* E154408A, one of these coding for an alkylglycerone-phosphate synthase (K00803; EC number 2.5.1.26; COG0277), which was not detected in any other *A. johnsonii* genome (Fig. 4B).

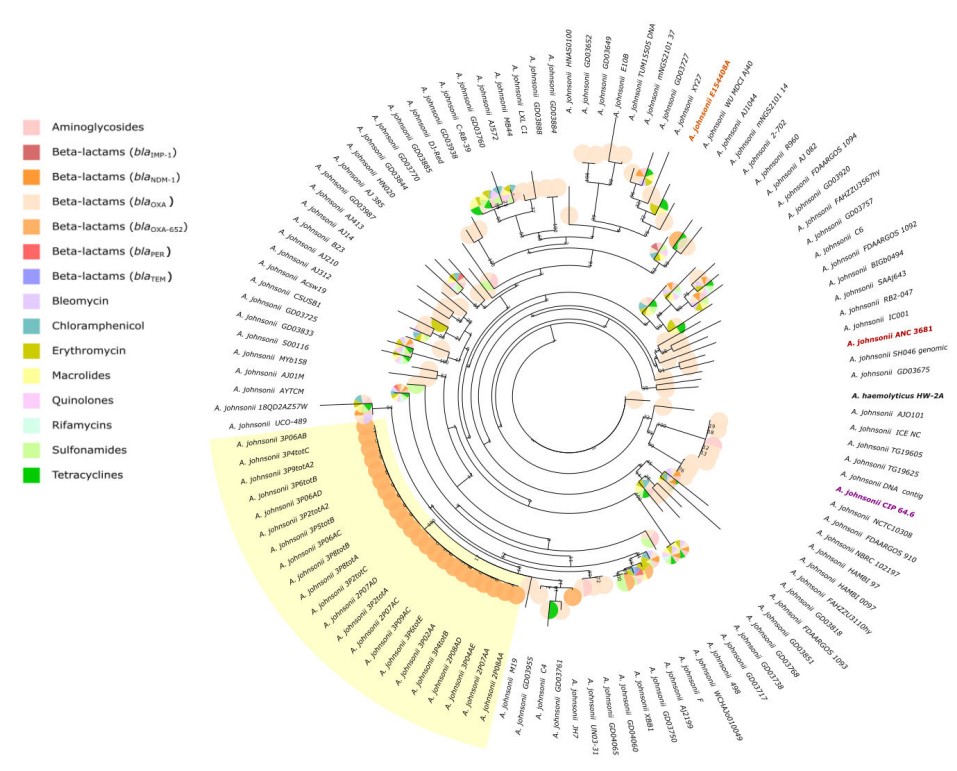

**FIG 3** Phylogenomic analysis. Phylogeny of PHSF-originating isolates (highlighted in yellow) and E154408A strain (orange) inferred through Maximum Likelihood. Pie charts associated with the terminal branches represent identified ARGs in corresponding genomes. The *A. johnsonii* reference genome ANC 3681 and the CIP 64.6$^T$ type strain are highlighted in red and purple, respectively. The reference genome of *A. haemolyticus* HW-2A served as an outgroup.

## Species-wide characterization of the plasmidome of *A. johnsonii*

A total of 89 distinct plasmids were predicted across all *A. johnsonii* genomes, including 33 newly described types. Of these, 51 were assigned a relaxase type (57.30%) (see Table S3 at https://doi.org/10.5281/zenodo.18473520). Genomes were identified to carry between 0 and 8 putative plasmids (mean, 1.83 per genome). AC600 was predicted with the highest frequency across the dataset, appearing in 19/112 (16.96%) of the genomes, followed by AC961 (14/112, 12.50%). While five PHSF isolates lacked detectable plasmids, the rest of the PHSF-derived isolates ($n = 17$) consistently encoded a shared set of two putative plasmids (AD642 and AC600) predicted to originate from *Acinetobacter wuhouensis* (34) and *A. johnsonii*, respectively. Notably, AC600 was found in 15/22 (68.19%) of the PHSF-derived *A. johnsonii* genomes and only four other *A. johnsonii* genomes from the broader dataset. Based on Mash distance neighbor analysis, AC600 is highly similar to a plasmid originating from isolate *A. johnsonii* XBB1 (mash distance 0.005). Two additional plasmids (AH350 and AD731), predicted to originate from *Moraxella osloensis* (35) and *A. baumannii* (36), were identified in 2P07AA and 3P06AC strains, respectively. The clinical strain E154408A encoded six distinct putative plasmid types, including AH350 previously found in one PHSF-originating isolate.

## Antimicrobial resistance

A comprehensive search for ARGs in all available genomes of *A. johnsonii* identified a species-wide resistome comprising 62 ARGs predicted to confer resistance to aminoglycosides [*aac*(3), *aac*(6′), *ant*(2″), *aph*(3′), *aph*(4), *aph*(6)-type genes], carbapenems [*bla*IMP-1, *bla*OXA, *bla*NDM-1, *bla*PER-1, *bla*PER-2, *bla*TEM-1, *BRP(MBL)*], macrolides (*ereA2*, *estT*,

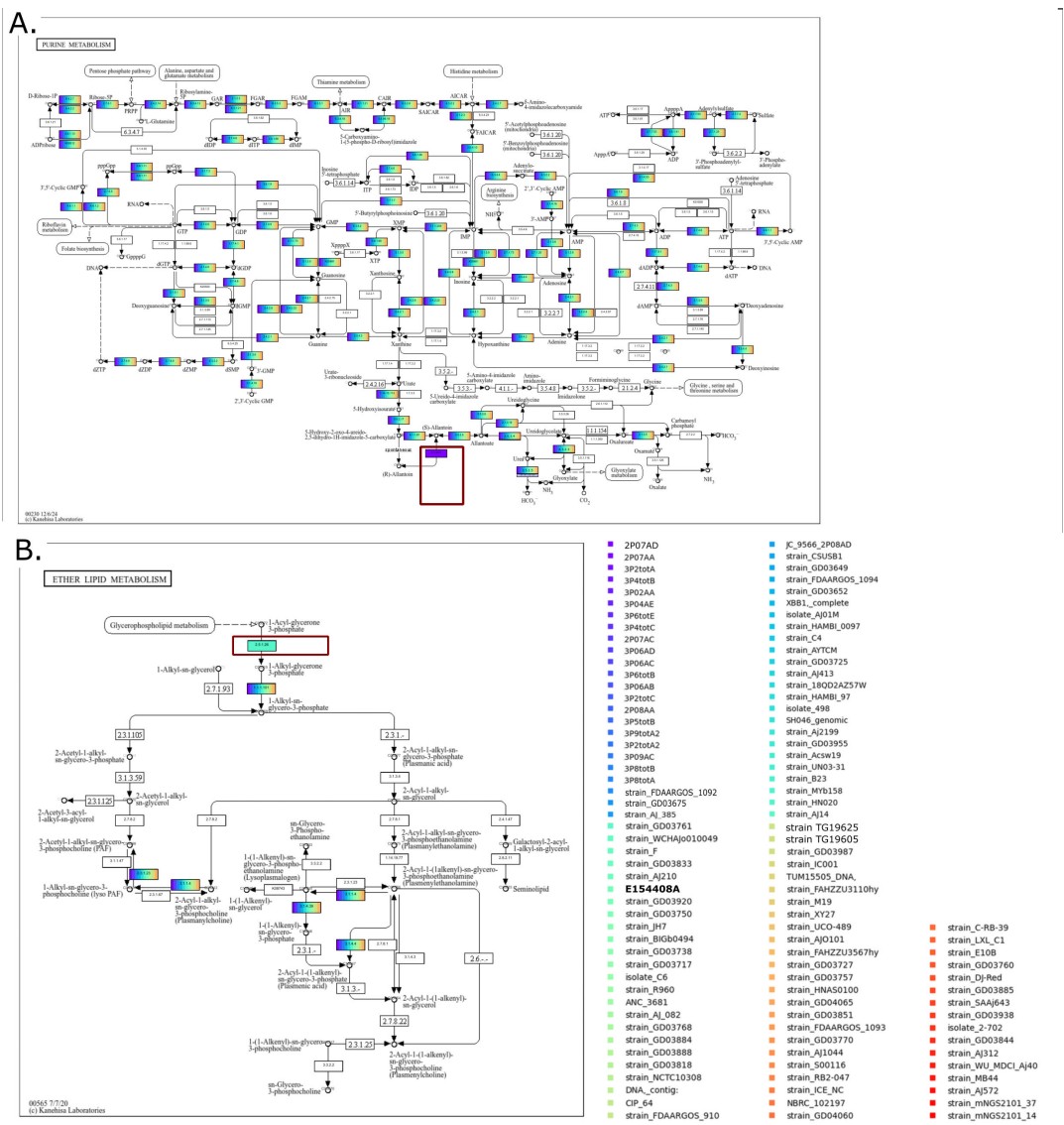

**FIG 4** Functional annotation of nucleic acid sequences and metabolic mapping. Genome-based metabolic profiling and pathway predictions represented using KEGGCharter. (A) PHSF-derived isolates encode a unique allantoin racemase (red square) involved in purine metabolism; (B) E154408A strain encodes an alkylglycerone-phosphate synthase involved in ether lipid metabolism, which was not detected in any other *A. johnsonii* genome.

*mphE*, and *msrE*), phenicols (*catB3*, *cmlB1,* and *floR*), quinolones (*qnrVC6*), sulfonamide (*sul1* and *sul2*), tetracycline (tet genes), and rifamycin *(arr-3)* (Fig. 5). Among these, 37 ARGs were identified on predicted plasmid sequences in 35 genomes of *A. johnsonii* (see Table S4 at https://doi.org/10.5281/zenodo.18473520). *A. johnsonii* AYTCM strain (human source, Zhejiang, China) was observed to encode the highest number of ARGs ($n = 19$), including multiple genes conferring resistance to carbapenems, namely $BRP(MBL)$, $bla_{IMP-1}$, $bla_{NDM-1}$, $bla_{OXA-58}$, $bla_{OXA-652}$, $bla_{PER-1}$. Of note, 100% (112/112) of *A. johnsonii* genomes were observed to carry at least one $bla_{OXA}$. One single ARG, $bla_{OXA-652}$, potentially conferring broad-spectrum resistance to β-lactams, was chromosomally detected in all PHSF-originating *A. johnsonii* genomes. $bla_{OXA-652}$ was also detected in *A. johnsonii* strains mNGS2101_14 (human source, Hangzhou, China), JH7 (environmental source, Guangxi, China), GD03750 (environmental source, Pakistan), F (environmental source, Taizhou, China), AYTCM (human source, Zhejiang, China), Acsw19 (environmental source, Luzhou, China), and XBB1 (environmental source, Chengdu, China). A total of four ARGs were chromosomally detected in the *A. johnsonii* E154408A strain, namely

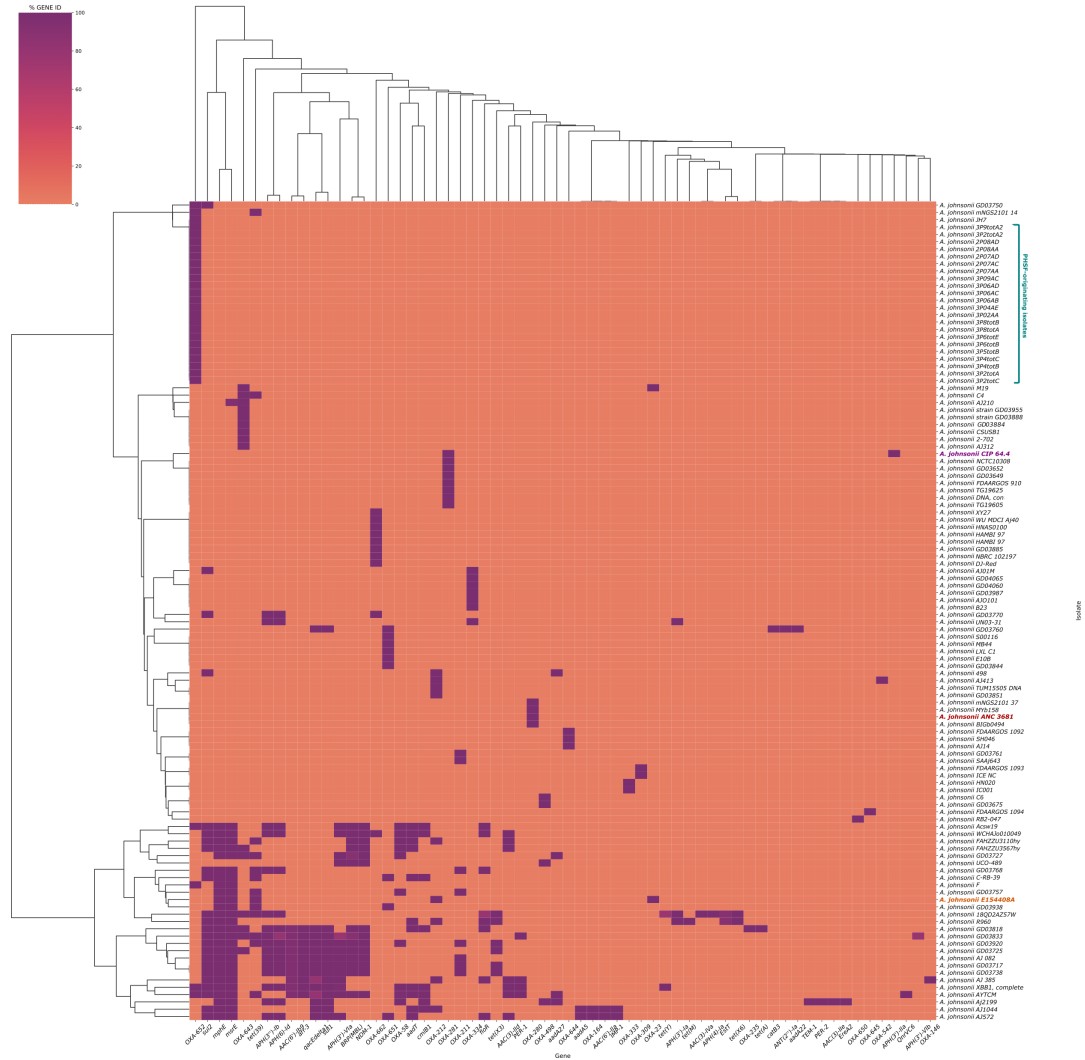

**FIG 5** Species-wide characterization of the resistome of *A. johnsonii*. Species-wide characterization of the resistome of *A. johnsonii* (n = 112 strains), including PHSF-originating genomes (teal) and *A. johnsonii* E154408A (orange). *A. johnsonii* reference genome and type strain are highlighted in red and purple, respectively.

bla$_{OXA-212}$, *mphE*, *msrE*, and *tet(39),* potentially conferring resistance to β-lactams, macrolides, sulfonamide, and tetracycline antibiotics. However, these ARGs were found in a variable number (range: 8–26) of other *A. johnsonii* genomes. Of note, a second oxacillinase gene, bla$_{OXA-23}$, was detected in E154408A and predicted to have plasmidic location (plasmid AD622). bla$_{OXA-23}$ was also predicted on the same plasmid in the *A. johnsonii* M19 strain.

Phenotypic AST revealed that E154408A and 8 out of the 14 tested PHSF-originating isolates (i.e., 3P2-tot-C, 3P5-tot-B, 3P6-tot-E, 3P8-tot-B, 3P04AE, 3P06AC, 3P06AD, and 3P09AC) showed intermediate susceptibility to cefotaxime, while the remaining isolates were sensitive (see Fig. S4 at https://doi.org/10.5281/zenodo.18473520). In addition, five isolates (i.e., 3P6-tot-E, 3P8-tot-B, 3P04AE, 3P06AD, and 3P09AA) demonstrated intermediate susceptibility to piperacillin, with the remaining isolates being susceptible. Notably, *A. johnsonii* E154408A exhibited multidrug phenotypic resistance to 6 out of the 12 tested antimicrobials, particularly to tetracycline, piperacillin, cefepime, and ciprofloxacin, as well as the carbapenems, imipenem and meropenem.

## Species-wide characterization of putative virulence-associated genes of *A. johnsonii*

A total of eight known VFs were identified across all available *A. johnsonii* genomes (see Fig. S3 at https://doi.org/10.5281/zenodo.18473520). *pilT* and *pilG* were ubiquitously found in 100% (112/112) of the genomes and *ompA* in 98.2% (110/112) of the genomes. PHSF-originating *A. johnsonii* genomes carried the same subset of six VFs, *ompA*, *pilG*, *pilT*, *hcp/tssD*, *tssC*, and *tse4*. Three VFs, *pilG*, *pilT*, and *ompA*, were identified in the E154408A strain.

## Species-wide characterization of antibacterial biocide and metal resistance gene profiles in *A. johnsonii*

A total of 146 antibacterial biocide and metal resistance genes were identified across the dataset using BacMet (Fig. 6). Highly prevalent genes, appearing in over 97% of the genomes used, included the AdeIJK efflux pump genes *adeI/J/K,* which mediate resistance to a broad range of antibiotics (including beta-lactams and fluoroquinolones) and anionic surfactant compounds; *oxyRkp* (resistance to hydrogen peroxide and detergents); *fabI* and *mexT* (triclosan); *sitB* and *sodB* (hydrogen peroxide); *rpoS* and *gadC/xasA* (hydrochloric acid); *evgA* (sodium deoxycholate); and multiple metal resistance genes.

Among the 22 PHSF-derived genomes, an average of 82 (range: 81–83) of the 146 identified genes were detected. Notably, *ssmE*, encoding the SsmE multidrug efflux pump, was present in 6/22 of the PHSF-derived genomes compared with 73.03% (65/112) of the remaining *A. johnsonii* isolates. Eighty-seven antibacterial biocide and metal resistance genes were identified in the clinical E154408A strain. While PHSF-originating isolates and E154408A carried the complete AdeIJK efflux pump gene cassette, PHSF genomes additionally possessed *adeA* and *adeB*, two components of the AdeABC multidrug resistance efflux pump also found in 43.75% of the rest of the dataset. Further screening with a custom database revealed the presence of *adeW* and *adeT* (homologs of *adeA*) and *adeR* and *adeS* (other components of AdeABC) in all PHSF isolates, though *adeS* showed reduced sequence identity relative to the reference sequence. The *adeN* repressor of AdeIJK was also detected in all PHSF-derived isolates, albeit with diminished identity to the reference.

## DISCUSSION

Taxonomic classification of both PHSF-originating isolates (*n* = 22) and the clinical strain E154408A as *A. johnsonii* was confirmed through a combination of *in vitro* and *in silico* approaches, using both the *A. johnsonii* reference genome and the type strain CIP 64.6$^{T}$ for comparison. Bootstrap-supported core pangenome-based phylogeny showed that the PHSF-derived isolates form a monophyletic clade diverging from the main *A. johnsonii* lineage, suggesting the potential emergence of genomic features favorable for adaptation to extremely clean environments. While observed 40–77 core SNP differences between the 22 PHSF-derived isolates are compatible with a single clonal lineage, this exceeds the threshold typically used to define outbreak-related strains in the neighboring species *A. baumannii* (≤10 SNPs) (37), which is consistent with a dataset of closely related yet genetically distinct isolates. Such divergence could plausibly reflect homologous recombination events, which introduced localized allelic replacements resulting in increased core SNP distances among the 22 PHSF-derived isolates without affecting their overall relatedness. Natural competence is indeed common across the *Acinetobacter* genus (38, 39), and previous literature suggested that *A. johnsonii* can also acquire exogenous DNA, directly impacting its enhanced adaptability to extreme environments (6).

Genome-based metabolic profiling and pathway predictions revealed only subtle differences between PHSF-originating and the remaining *A. johnsonii* genomes, indicating the absence of broad specialized pathways conferring adaptation to

cleanroom conditions. Nonetheless, the PHSF-derived isolates uniquely encode an allantoin racemase gene absent in all other *A. johnsonii* genomes examined. Allantoin racemase catalyzes the interconversion of R- and S-allantoin, potentially enhancing the breakdown of purines into metabolically useful intermediates (40). Enhanced purine utilization could offer a nutritional edge in nutrient-poor environments by providing an additional route for nitrogen and carbon acquisition (40). This hypothesis remains, however, highly speculative, and further research is needed to determine whether this unique and highly conserved trait contributes to more efficient nutrient utilization and, ultimately, to the survival of *A. johnsonii* in the cleanroom environment.

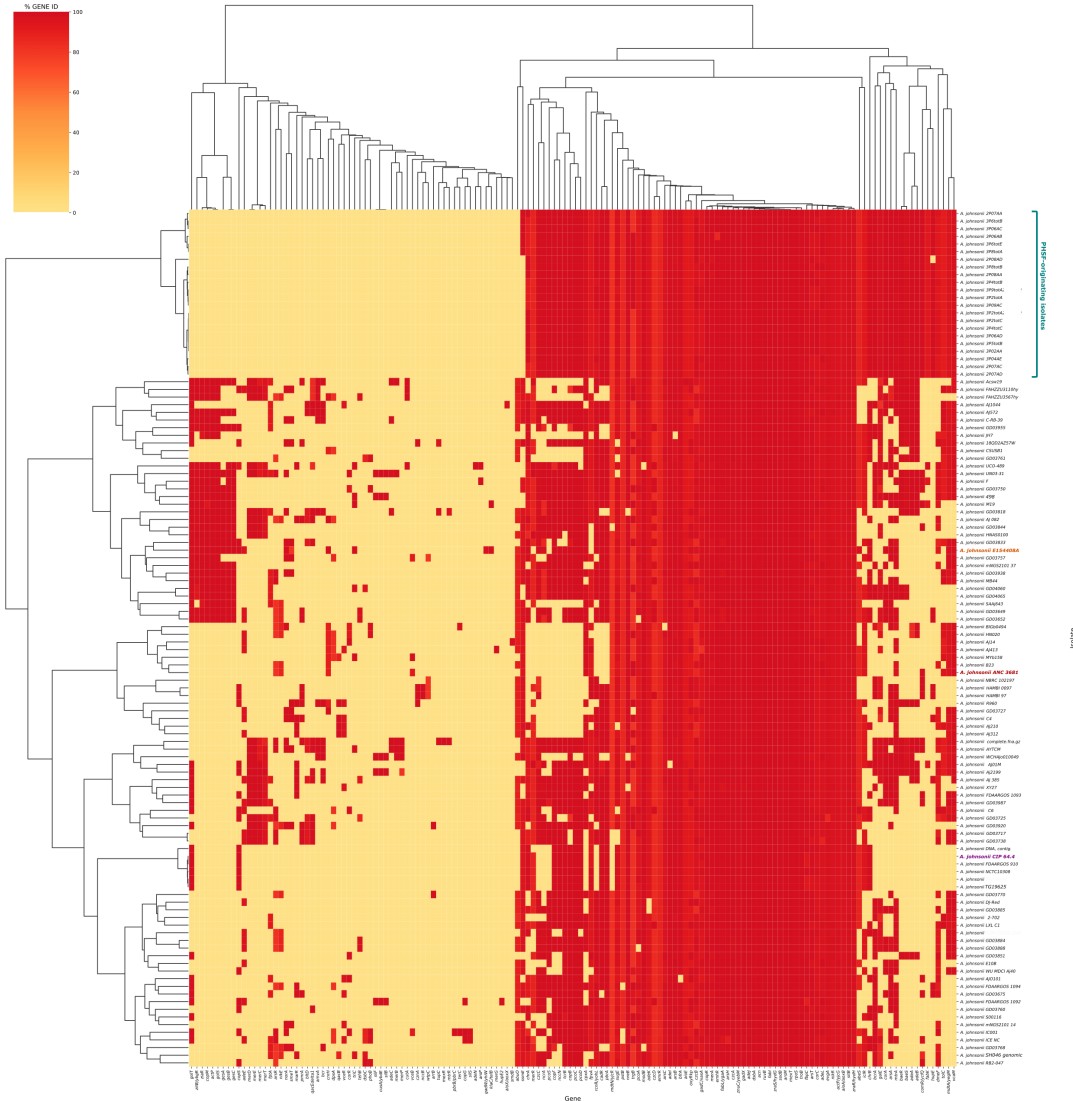

**FIG 6** Species-wide characterization of antibacterial biocide and metal resistance gene profiles in *A. johnsonii*. Species-wide characterization of the profile of antibacterial biocide and metal resistance genes using BacMet in *A. johnsonii* (*n* = 112 strains), including PHSF-originating genomes (teal) and *A. johnsonii* E154408A (orange). *A. johnsonii* reference genome and type strain are highlighted in red and purple, respectively. A total of 146 antibacterial biocide and metal resistance genes were identified across the dataset. Highly prevalent genes, appearing in over 97% of the genomes used, included *adeI/J/K*, which mediate resistance to a broad range of antibiotics (including beta-lactams and fluoroquinolones) and anionic surfactant compounds; *oxyRkp* (resistance to hydrogen peroxide and detergents); *fabI* and *mexT* (triclosan resistance); *sitB* and *sodB* (hydrogen peroxide); *rpoS* and *gadC/xasA* (hydrochloric acid); *evgA* (sodium deoxycholate); and multiple genes conferring resistance to arsenic, cadmium, cobalt, copper, iron, magnesium, mercury, and zinc. An average of 82 (range: 81–83) of the 146 identified genes were detected among the 22 PHSF-derived genomes, while 87 were identified in the clinical E154408A strain. While PHSF-originating isolates and E154408A carried the complete AdeIJK efflux pump gene cassette, the PHSF genome additionally possessed *adeA* and *adeB*, two components of the AdeABC multidrug resistance efflux pump also found in 43.75% of the rest of the dataset.

The number of predicted plasmids per PHSF-derived genome varied from zero to three, suggesting dynamic events of plasmid gain and/or loss across the monophyletic population and further supporting the genetic diversification of PHSF-originating isolates. Notably, the nearest neighbor of one of the two mobilizable plasmids identified in E154408A (AC935) is known to carry $bla_{OXA}$ genes (41), indicating a potential pathway for the acquisition of $bla_{OXA-23}$.

Due to the contig-level assembly status of these genomes, it remains difficult to determine conclusively whether resistance and putative virulence-associated genes identified in these isolates are plasmid-encoded or located on chromosomal regions. Further high-resolution sequencing and functional studies would be necessary to clarify their genomic origins.

Species-wide characterization of the *A. johnsonii* resistome revealed the intrinsic presence of $bla_{OXA}$ in all examined genomes, with no exceptions. This is consistent with previous reports that *A. johnsonii* can persist in polluted environments containing antibiotic residues and highlights the risk posed by *Acinetobacter* species in causing difficult-to-treat infections due to limited therapeutic options (42). While the species-wide presence of $bla_{OXA}$ in every *A. johnsonii* genome is not common knowledge, our findings provide a large-scale confirmation of $bla_{OXA}$ ubiquity in *A. johnsonii*.

All PHSF-originating isolates consistently carried a single chromosomal ARG, namely $bla_{OXA-652}$ from the $bla_{OXA-211}$ family, known for its narrow-spectrum cephalosporin hydrolysis and lack of clinically significant carbapenem resistance (43). Indeed, none of the PHSF-originating isolates exhibited carbapenem resistance at the phenotypic level. In contrast, E154408A displayed phenotypic resistance to carbapenems (imipenem and meropenem), cefepime, ciprofloxacin, piperacillin, and tetracyclines, consistent with its genomic profile marked by *mphE*, *msrE*, *tet*(39), and by the co-occurrence of the chromosomal $bla_{OXA-212}$ and plasmidic $bla_{OXA-23}$. Although biosynthesis of these oxacillinases was not directly verified, the observed carbapenem-resistant phenotype is consistent with the presence of $bla_{OXA-23}$ (42). However, resistance of the E154408A strain to carbapenems may also reflect an additive resistance phenotype due to the cumulative presence of intrinsic and acquired oxacillinases, which elevate antimicrobial resistance levels beyond that expected from a single enzyme alone (44). Collectively, these findings highlight *A. johnsonii* as an understudied yet emerging antimicrobial resistance threat and suggest the potential ability of this species to develop more extensive antimicrobial resistance profiles due to the combined presence of chromosomal and acquired oxacillinases.

We note, however, the lack of clinically certified phenotypic AST assays tailored for *A. johnsonii*'s specific optimal growth conditions and the absence of guidelines (e.g., CLSI and EUCAST) to interpret results accordingly. Addressing this gap is important for managing *A. johnsonii* with potential MDR phenotype in clinical decision-making.

The widespread presence of key components of the Type VI Secretion System (T6SS) (*tse4*, *tssC*, and *hcp/tssD*) among PHSF-derived isolates, less commonly found in the broader dataset, aligns with evidence that a more complete T6SS pathway may assist their tolerance to nutrient-scarce environments. This is supported by previous literature showing how strict cleaning procedures can promote selection for functions associated with membrane transport and secretion, including T6SS, to gather nutrients from highly competitive, nutrient-poor environments (45). Additionally, the fact that PHSF-originating genomes belong to the subset of *A. johnsonii* strains encoding both AdeIJK and a nearly complete (missing only *adeC*) AdeABC efflux pumps may contribute to explaining their resilience to repeated decontamination protocols, as supported by examples of increased tolerance of benzalkonium chloride, a decontamination reagent widely used in NASA cleanrooms, associated with the presence of AdeABC and AdeIJK efflux pumps in *A. johnsonii* (46). Notably, previous studies demonstrated the activity of AdeABC despite the absence of *adeC*, compensated by *adeK* (47).

Genome-to-metagenome mapping revealed the presence in PHSF of genomic signatures and a minimal population of viable cells of *A. johnsonii* over 10 years

post-initial isolation and overall greater bacterial load in KSC-PHSF compared to JPL-SAF, possibly due to the stricter disinfection regime applied in JPL-SAF following spacecraft assembly activities in 2018 (30).

In conclusion, our results indicate deviation of PHSF-derived isolates from the main *A. johnsonii* species lineage and the emergence in their genomes of unique and conserved traits that may explain adaptation of this non-spore-forming microorganism to extremely clean and nutrient-scarce environments. Further research, however, is required to demonstrate whether these genomic traits contributed to the survival of *A. johnsonii* in NASA cleanrooms. Through the comprehensive genomic characterization of *A. johnsonii*'s resistome, our results provide a valuable, large-scale confirmation of intrinsic *bla*$_{OXA}$ ubiquity in this species. With the documentation of the first reported carbapenem-resistant patient colonization case in Ireland and Europe, our findings also shed light on the potential of *A. johnsonii* to develop enhanced antimicrobial-resistant phenotypes and the risks related to the persistence of this microbial contaminant as a reservoir and vector of AMR in controlled environments. Collectively, our findings will help assess and manage the contamination risk of *A. johnsonii* across clinical, terrestrial, and extraterrestrial settings.

## ACKNOWLEDGMENTS

This work was supported by Prof. A.S.R.'s KAUST Baseline Grant (BAS/1/1096-01-01). Part of the research described in this publication was carried out at the Jet Propulsion Laboratory, California Institute of Technology, under a contract with the National Aeronautics and Space Administration. This research was funded by a 2012 Space Biology NNH12ZTT001N grant no. 19-12829-26 under Task Order NNN13D111T award to K.V. A.T. is supported by the Environmental Protection Agency Research Programme 2021-2030 as a Government of Ireland initiative funded by the Department of the Environment, Climate and Communications. P.S. is supported through the Prime Minister's Research Fellowship from the Ministry of Education, Government of India. F.M. is supported by Taighde Éireann–Research Ireland under Grant number GOIPG/2023/4515. The funders had no role in study design, data collection and interpretation, the writing of the manuscript, or the decision to submit the work for publication.

We thank Patrick Driguez, Angel Angelov, and Alexander Putra at KAUST Core Laboratories, King Abdullah University of Science and Technology (KAUST), Thuwal, Kingdom of Saudi Arabia, for sequencing the PHSF-originating isolates. We thank Tahira Jamil at Biological and Environmental Sciences and Engineering Division (BESE), King Abdullah University of Science and Technology (KAUST), Thuwal, Kingdom of Saudi Arabia, for assisting with genome assembly. We thank the staff of Galway Reference Laboratory Services, Galway University Hospital, Galway, Ireland, for reporting and sequencing the clinical originating isolate.

G.M. and K.V. conceptualized the study; G.M. and K.V. designed the methodology; A.T., G.M., V.V., and P.S. helped with software; A.T., G.M., A.O'C., V.V., and P.S. conducted formal analysis; A.T., G.M., A.O'C., F.M., A.K., and B.H. conducted the investigation; G.M., C.C., A.S.R., K.R., and K.V. provided resources; A.T. and G.M. curated the data; A.T., A.O'C., V.V., P.S., and A.K. wrote the original draft; A.T., G.M., N.K.S., and K.V. reviewed and edited the manuscript; A.T. and V.V. visualized the study; G.M., K.R., and K.V. supervised the study; A.T., G.M., and K.V. contributed to project administration; G.M. acquired funding.

## AUTHOR AFFILIATIONS

[1]Antimicrobial Resistance and Microbial Ecology Group, School of Medicine, University of Galway, Galway, Ireland

[2]Department of Biotechnology, Bhupat and Jyoti Mehta School of Biosciences, Indian Institute of Technology (IIT) Madras, Chennai, India

[3]Centre for Integrative Biology and Systems mEdicine (IBSE), Indian Institute of Technology (IIT) Madras, Chennai, India

[4]Wadhwani School of Data Science and AI (WSAI), Indian Institute of Technology (IIT) Madras, Chennai, India

[5]Galway University Hospital, Galway, Ireland

[6]Atlantic Technological University, Galway, Ireland

[7]NASA Jet Propulsion Laboratory (JPL), California Institute of Technology, Pasadena, California, USA

[8]Biological and Environmental Sciences and Engineering Division (BESE), King Abdullah University of Science and Technology (KAUST), Thuwal, Kingdom of Saudi Arabia

[9]Department of Data Science and AI, Wadhwani School of Data Science and AI (WSAI), Indian Institute of Technology (IIT) Madras, Chennai, India

## AUTHOR ORCIDs

Anna Tumeo http://orcid.org/0009-0007-7689-2809

Georgios Miliotis http://orcid.org/0000-0002-0944-2206

Andy O'Connor http://orcid.org/0009-0002-4510-8915

Varsha Vijayakumar http://orcid.org/0009-0000-1184-7862

Pratyay Sengupta http://orcid.org/0000-0002-0184-9335

Francesca McDonagh http://orcid.org/0000-0002-0830-4899

Aneta Kovarova http://orcid.org/0009-0008-4812-1553

Brigid Hooban http://orcid.org/0000-0003-4882-6891

Nitin Kumar Singh http://orcid.org/0000-0001-5344-1190

Alexandre Soares Rosado http://orcid.org/0000-0001-5135-1394

Karthik Raman http://orcid.org/0000-0002-9311-7093

Kasthuri Venkateswaran http://orcid.org/0000-0002-6742-0873

## FUNDING

| Funder | Grant(s) | Author(s) |
|---|---|---|
| King Abdullah University of Science and Technology (KAUST) | BAS/1/1096-01-01 | Alexandre Soares Rosado |
| National Aeronautics and Space Administration (NASA) | 19-12829-26 | Kasthuri Venkateswaran |
| Environmental Protection Agency (EPA) | 2022-HE-1145 | Georgios Miliotis |
| Taighde Eireann - Research Ireland | GOIPG/2023/4515 | Georgios Miliotis |
| Ministry of Education, Government of India | | Karthik Raman |

## AUTHOR CONTRIBUTIONS

Anna Tumeo, Data curation, Formal analysis, Investigation, Project administration, Software, Visualization, Writing – original draft, Writing – review and editing | Georgios Miliotis, Conceptualization, Data curation, Formal analysis, Funding acquisition, Investigation, Methodology, Project administration, Resources, Software, Supervision, Writing – review and editing | Andy O'Connor, Formal analysis, Investigation, Writing – original draft | Varsha Vijayakumar, Formal analysis, Software, Visualization, Writing – original draft | Pratyay Sengupta, Formal analysis, Software, Writing – original draft | Francesca McDonagh, Investigation | Aneta Kovarova, Investigation, Writing – original draft | Christina Clarke, Resources | Brigid Hooban, Investigation | Nitin Kumar Singh, Writing – review and editing | Alexandre Soares Rosado, Resources | Karthik Raman, Resources, Supervision | Kasthuri Venkateswaran, Conceptualization, Methodology, Project administration, Resources, Supervision, Writing – review and editing

## DATA AVAILABILITY

The genomes described in this project have been deposited at DDBJ/ENA/GenBank under the BioProject numbers PRJNA1128436 and PRJNA1209713. All supplemental material can be found at https://doi.org/10.5281/zenodo.18473520.

## ADDITIONAL FILES

The following material is available online.

### Open Peer Review

**PEER REVIEW HISTORY (review-history.pdf).** An accounting of the reviewer comments and feedback.

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
