## [Reviewer comments · Microbiology Spectrum]

Microbiology Spectrum

Plasmidome, resistome, and virulence-associated genes characterization of *Acinetobacter johnsonii* in NASA cleanrooms and a clinical setting.

Anna Tumeo, Georgios Miliotis, Andy O' Connor, Varsha Vijayakumar, Pratyay Sengupta, Francesca McDonagh, Aneta Kovarova, Christina Clarke, Brigid Hooban, Nitin Kumar Singh, Alexandre Rosado, Karthik Raman, and Kasthuri Venkateswaran

Corresponding Author(s): Karthik Raman, Indian Institute of Technology Madras

Review Timeline:

Submission Date:	August 13, 2025
Editorial Decision:	December 26, 2025
Revision Received:	January 19, 2026
Accepted:	January 23, 2026

Editor: Ayesha Khan

Reviewer(s): Disclosure of reviewer identity is with reference to reviewer comments included in decision letter(s). The following individuals involved in review of your submission have agreed to reveal their identity: Phoebe Lostroh (Reviewer #1)

Transaction Report:

DOI: <https://doi.org/10.1128/spectrum.02503-25>

Re: Spectrum02503-25 (Plasmidome, resistome, and virulence-associated genes characterization of *Acinetobacter johnsonii* in NASA cleanrooms and a clinical setting.)

Dear Dr. Karthik Raman:

Thank you for the privilege of reviewing your work. Below you will find my comments, instructions from the Spectrum editorial office, and the reviewer comments.

Revision Guidelines

Sincerely,
Ayesha Khan
Editor
Microbiology Spectrum

Reviewer #1 (Comments for the Author):

- Significance to the target scientific community
 - o Important for understanding the microbiology of extreme clean rooms, for understanding reservoirs of antimicrobial resistance genes and also astrobiologists and microbiologists who study life in extreme environments
- Originality

- o The authors investigate a unique set of difficult-to-obtain samples.
- Appropriateness of the approach or experimental design
- o The combination of culture followed by genome sequencing and comparison to other genomes is appropriate for the experimental design
- Adequacy of experimental techniques
- o The authors should add a few sentences about the isolation culture procedure. (lines 108-112)
- o Genome analysis experimental techniques are described in sufficient detail.
- Soundness of conclusions and interpretation
- o Reasonable approach to *A. jo* genome analysis using an interesting data subset from NASA clean rooms. They also include genomes from previously-detected *A. jo* DNA or cells. The conclusions about the presence of ARGs, plasmids, VFs, and metal/biocide resistance genes are supported by the comparative data.
- o The authors have not discussed how natural transformation may have affected the core SNP clusters from the PHSF isolates. Are *A. jo* competent?
- Relevance of discussion
- o The discussion is relevant to *Acinetobacter* investigators and microbiologists investigating reservoirs of antimicrobial resistance genes. The ideas about enhanced purine utilization are highly speculative.
- o Discussion of T6SS and their use to obtain DNA by killing other cells could enhance the manuscript. Both T6SS and TFP are expressed in stressful conditions in other *Acinetobacter* spp
- Adequacy of title and abstract
- o Why has the antibiotic resistome been omitted from the title which otherwise summarizes the main thrust of the investigation?
- Appropriateness of figures and tables
- o Figures 1-5 and Table 1 should be included in print; Figure 5 could appear earlier in the manuscript.
- Appropriateness of supplemental material intended for posting (if applicable)
- o The supplemental materials included should be accessible to other investigators who have read this paper.
- Whether it describes misuse of microbial systems or the information derived therefrom
- o It does not.

In response to Reviewer #1 (Comments for the Author)

- **Significance to the target scientific community**

- o Important for understanding the microbiology of extreme clean rooms, for understanding reservoirs of antimicrobial resistance genes and also astrobiologists and microbiologists who study life in extreme environments

- **Originality**

- o The authors investigate a unique set of difficult-to-obtain samples.

- **Appropriateness of the approach or experimental design**

- o The combination of culture followed by genome sequencing and comparison to other genomes is appropriate for the experimental design

- **Adequacy of experimental techniques**

- o The authors should add a few sentences about the isolation culture procedure. (lines 108-112). **Response:** Relevant details were added as recommended (lines 109-129).

- o Genome analysis experimental techniques are described in sufficient detail.

- **Soundness of conclusions and interpretation**

- o Reasonable approach to *A. jo* genome analysis using an interesting data subset from NASA clean rooms. They also include genomes from previously-detected *A. jo* DNA or cells. The conclusions about the presence of ARGs, plasmids, VFs, and metal/biocide resistance genes are supported by the comparative data.

- o The authors have not discussed how natural transformation may have affected the core SNP clusters from the PHSF isolates. Are *A. jo* competent? **Response:** The following statements discussed in the text will clarify this comment: “*Such divergence could plausibly reflect homologous recombination events, which introduced localized allelic replacements resulting in increased core SNP distances among the 22 PHSF-derived isolates without affecting their overall relatedness*”; “*Natural competence is indeed common across the Acinetobacter genus, and previous literature suggested that A. johnsonii can also acquire exogenous DNA, directly impacting on its enhanced adaptability to extreme environments*” (lines 362-367).

- **Relevance of discussion**

- o The discussion is relevant to *Acinetobacter* investigators and microbiologists investigating reservoirs of antimicrobial resistance genes. The ideas about enhanced purine utilization are highly speculative. **Response:** As suggested by the reviewer, we have toned down the speculation since more research is needed (lines 375-376).

- o Discussion of T6SS and their use to obtain DNA by killing other cells could enhance the manuscript. Both T6SS and TFP are expressed in stressful conditions in other *Acinetobacter* spp. **Response:** Reviewer’s suggestion about T6SS expression is now included in the discussion as follows: “*This is supported by previous literature showing how strict cleaning procedures can promote selection for functions associated with membrane transport and secretion, including T6SS, to gather nutrients from highly competitive, nutrient-poor environments*” (lines 417-420).

- **Adequacy of title and abstract**

- o Why has the antibiotic resistome been omitted from the title which otherwise summarizes the main thrust of the investigation? **Response:** “Resistome” is included in the title as follows: “*Plasmidome, resistome, and virulence-associated genes characterization...*”.

- **Appropriateness of figures and tables**

- o Figures 1-5 and Table 1 should be included in print; Figure 5 could appear earlier in the manuscript. **Response:** Old Figure 5 is now presented as Figure 1 of the manuscript. Remaining figures were adjusted accordingly.

- **Appropriateness of supplemental material intended for posting (if applicable)**

- o The supplemental materials included should be accessible to other investigators who have read this paper.

- **Whether it describes misuse of microbial systems or the information derived therefrom**

- o It does not.

Re: Spectrum02503-25R1 (Plasmidome, resistome, and virulence-associated genes characterization of *Acinetobacter johnsonii* in NASA cleanrooms and a clinical setting.)

Dear Dr. Karthik Raman:

Your manuscript has been accepted, and I am forwarding it to the ASM production staff for publication. Your paper will first be checked to make sure all elements meet the technical requirements. ASM staff will contact you if anything needs to be revised before copyediting and production can begin. Otherwise, you will be notified when your proofs are ready to be viewed.

Sincerely,
Ayesha Khan
Editor
Microbiology Spectrum